# Preliminary observations on the mandibles of palaemonoid shrimp (Crustacea: Decapoda: Caridea: Palaemonoidea)

Christopher W. Ashelby[1,3], Sammy De Grave[2] and Magnus L. Johnson[3]

[1] APEM Ltd., 7 Diamond Centre, Letchworth Garden City, United Kingdom
[2] Oxford University Museum of Natural History, Oxford, United Kingdom
[3] CEMS, University of Hull, Scarborough Campus, Scarborough, United Kingdom

## ABSTRACT

The mandibles of caridean shrimps have been widely studied in the taxonomy and functional biology of the group. Within the Palaemonoidea the mandibles reach a high level of structural diversity reflecting the diverse lifestyles within the superfamily. However, the majority of studies have been restricted to light microscopy, with the ultrastructure at finer levels poorly known. This study investigates the mandible of nine species belonging to six of the recognised families of the Palaemonoidea using SEM and analyses the results in a phylogenetic and dietary framework. The results of the study indicate that little phylogenetic information is conveyed by the structure of the mandible, but that its form is influenced by primary food sources of each species. With the exception of *Anchistioides antiguensis*, all species examined possessed cuticular structures at the distal end of the *pars molaris* (molar process). Five types of cuticular structures are recognised herein, each with a unique form, but variable in number, placement and arrangement. Each type is presumed to have a different function which is likewise related to diet.

## INTRODUCTION

Decapod crustaceans display a wide variety of modified mouthparts that serve both mechanical and sensory functions and have attracted the attention of taxonomists, systematicists and functional biologists for decades (e.g., *Borradaile, 1917*; *Fujino & Miyake, 1968*; *Roberts, 1968*; *Caine, 1975*; *Coombs & Allen, 1978*; *Schembri, 1982*; *Felgenhauer & Abele, 1985*; *Garm & Høeg, 2001*; *Garm, Hallberg & Høeg, 2003*; *Garm, 2004*). The semi-rigid, robust mandible has usually been attributed a solely mechanical function in the breaking down of food prior to ingestion, but a recent study of larval *Palaemon elegans Rathke, 1837* demonstrated that it possesses a variety of sensilla (*Geiselbrecht & Melzer, 2013*), suggesting that it may be more complex than previously thought. Indeed, *Borradaile (1917)* in his pioneering work on the structure and function of the mouthparts of palaemonid prawns concluded that "the mandible of the Crustacea is an exceedingly complicated, varied and

Corresponding author
Christopher W. Ashelby,
c.ashelby@apemltd.co.uk

interesting organ, presenting many problems and worthy of a great deal more attention than it has received." Nearly a century on and the caridean mandible, although superficially described in numerous taxonomic works, remains poorly studied at a structural level and very few studies have focussed on the detailed morphology and potential evolutionary drivers in relation to the form of the mandible. Recent investigations have added to our knowledge of the mandible across a range of crustacean taxa but have largely focussed on larvae (e.g., *Heral & Saudray, 1979*; *Casanova, De Jong & Moreau, 2002*; *Tziouveli, Bastos-Gomez & Bellwood, 2011*; *Geiselbrecht & Melzer, 2013*) or are restricted to a single or a small number of species within a single genus or family (e.g., *Fujino & Miyake, 1968*; *Caine, 1975*; *Coombs & Allen, 1978*; *Mielke, 1984*; *Felgenhauer & Abele, 1985*; *Hobbs, 1991*; *Moore, Rainbow & Larson, 1993*; *Richter, 2004*; *Arndt, Berge & Brandt, 2005*; *Mekhanikova, 2010*). Within the Palaemonoidea, the two most extensive studies on mandibles focus on the genus *Palaemon*, using light microscopy to examine its structure and function (*Borradaile, 1917*—as *Leander*) and interspecific variation (*Fujino & Miyake, 1968*).

Within the infraorder Caridea, the mandible is variously developed (*Burukovsky, 1986*) but is frequently comprised of a *pars incisivus* (incisor process) and *pars molaris* (molar process) and may be provided with a palp or not. Both the *pars incisivus* and the *pars molaris* are variable in form ranging from truncated to elongate, straight to markedly curved, narrow to flared, widely separated to barely separated and many gradations in between (*Burukovsky, 1986*). The distal portions of both processes are often provided with acute or rounded lobes ('teeth') or ridges but may be flattened. Either the *pars incisivus* or the *pars molaris* may be reduced or absent or they may be fused together. Due to this diversity in the development and form, features of the mandible have been used in the taxonomy of caridean shrimps, particularly in families where few characters exist to differentiate genera and species, such as in Palaemonidae. Additionally, several classifications of the Caridea have, in part, also been underpinned by features of the mandible (*Thompson, 1967*; *Christoffersen, 1990*; *Chace, 1992*).

In many decapods, mastication largely occurs in the gastric mill (*Caine, 1975*). *Patwardhan (1934)* expressed an opinion that many carideans lack a complex gastric mill and thus the mouthparts are correspondingly more developed, although more recent studies (e.g., *Felgenhauer & Abele, 1983*) demonstrate the presence of a gastric mill in a number of caridean families. Regardless, the mandible is involved in the initial breakdown of food and therefore has a large functional significance and thus its form may provide insights into the diet or feeding mode of the species. Indeed, species that have particular dietary regimes or feeding mechanisms tend to have correspondingly specialised mouthparts (*Caine, 1975*). During feeding, the *pars incisivus* is believed to be mostly used in cutting and slicing of food particles into more manageable portions whilst the *pars molaris* is usually thought to have a grinding function (*Bauer, 2004*), although *Felgenhauer & Abele (1985)* found that the mandible of atyid shrimps, that do possess a gastric mill, was not used for crushing food.

Whilst previous studies on shrimps have investigated mouthpart morphology of a single genus or species (*Borradaile, 1917*; *Fujino & Miyake, 1968*) or between genera

belonging to the same family (*Felgenhauer & Abele, 1985*), only the study of *Storch, Bluhm & Arntz (2001)* on three Antarctic shrimps has used SEM to investigate differences across families. The present, SEM-based, study was conceived to investigate the ultrastructure of the mandible in nine species belonging to nine different genera, across six out of seven families from the superfamily Palaemonoidea, thus covering a diversity of form and ecology, to evaluate the potential phylogenetic significance within the superfamily and the relationship between diet and structure.

## MATERIAL AND METHODS

*De Grave & Fransen (2011)* listed eight families included within the superfamily Palaemonoidea with the Palaemonidae further split into two subfamilies: the Palae-moninae and the Pontoniinae. However, the family Kakaducarididae has been recently synonymised with the Palaemonidae (see *Short, Humphrey & Page, 2013*) leaving seven valid families. Three of these families are monogeneric (Anchistioididae, Desmocarididae and Typhlocarididae) whilst the greatest diversity of both morphology and lifestyle is found in the subfamily Pontoniinae. No members of the Typhlocarididae were available for destructive examination via SEM and references to the morphology of the mandible in *Typhlocaris* are based on descriptions in the literature (*Calman, 1909*; *Parisi, 1921*; *Caroli, 1923*; *Caroli, 1924*; *Tsurnamal, 2008*). Despite several attempts to process left mandibles of *Euryrhynchus*, none survived the sonication stage intact and therefore observations are based on the right mandible only for this species. All specimens studied are held in the Zoological Collection of the Oxford University Museum of Natural History (OUMNH.ZC), with details included in Table 1.

The methods used for preparation of tissue follow those established by *Martin, Liu & Striley (2007)* and *De Grave & Goulding (2011)*. Mandibles were carefully dissected from specimens stored in 75% ethanol. After removal mandibles were passed through a graded ethanol series to distilled water, subjected to brief (5–15 s) sonication using a light surfactant, then re-hydrated in graded ethanol to 100%, with drying done via the HMDS (hexamethyldisilazane) method. Dried specimens were coated with a gold-palladium mixture in a Polaron E5000 coating unit and observed in a JEOL JSM-5510 microscope.

Terminology of the teeth on the *pars molaris* refers to their position *in situ* (see *Fujino & Miyake, 1968*), with setal definitions following *Garm (2004)*.

## RESULTS

Salient features of each mandible structure are outlined in Tables 2–5 and illustrated in Figures 1–7; only comparative remarks are detailed below.

The most common form of mandible of those species studied is bipartite, with a well developed *pars incisivus* and *pars molaris* (Table 2). Only in *Hymenocera picta Dana, 1852* (Tables 2 and 3; Fig. 4D) is the *pars incisivus* absent whilst in *Gnathophyllum elegans* (*Risso, 1816*) (Tables 2 and 3; Fig. 5A) it is reduced to a vestigial process. In all other species the structure of the *pars incisivus* is similar (Table 3) being flattened and provided with teeth distally. In *Pontonia pinnophylax* (*Otto, 1821*), a series of denticles is also present along the posterior margin (Table 3; Figs. 3A and 3C).

**Table 1** Species and museum accession numbers of specimens examined via SEM in this study.

| Species | Accession number |
|---|---|
| **Family Palaemonidae** | |
| **Subfamily Palaemoninae** | |
| *Palaemon macrodactylus Rathbun, 1902* | OUMNH.ZC 2006-01-0039 |
| *Macrobrachium nipponense* (*De Haan, 1849* (in De Haan, 1833–1850)) | OUMNH.ZC 2012-01-0060 |
| **Subfamily Pontoniinae** | |
| *Pontonia pinnophylax* (*Otto, 1821*) | OUMNH.ZC 2008-11-0081 |
| *Periclimenaeus caraibicus Holthuis, 1951* | OUMNH.ZC 2009-01-0101 |
| **Gnathophyllidae** | |
| *Gnathophyllum elegans* (*Risso, 1816*) | OUMNH.ZC 2011-09-0005 |
| **Hymenoceridae** | |
| *Hymenocera picta Dana, 1852* | OUMNH.ZC 2010-04-0017 |
| **Desmocarididae** | |
| *Desmocaris bislineata Powell, 1977* | OUMNH.ZC 2009-19-0001 |
| **Euryrhynchidae** | |
| *Euryrhynchus wrzesniowskii Miers, 1877* | OUMNH.ZC 2006-21-0001 |
| **Anchistioididae** | |
| *Anchistioides antiguensis Schmitt, 1924* | OUMNH.ZC 2007-14-0001 |

**Table 2** Summary of the features of the mandibles examined in this study.

| | *Pars molaris* | *Pars incisivus* | *Cuticular structures* | *Mandibular palp* |
|---|---|---|---|---|
| *Palaemon macrodactylus* | + | + | Type I | + |
| *Macrobrachium nipponense* | + | + | Type I | + |
| *Pontonia pinnophylax* | + | + | Type I | − |
| *Periclimenaeus caraibicus* | + | + | Type II | − |
| *Gnathophyllumelegans* | + | +/v | Type III | − |
| *Hymenocera picta* | + | − | Type IV | − |
| *Desmocaris bislineata* | + | + | Type V | − |
| *Euryrhynchus wrzesniowskii* | + | + | Type I | − |
| *Anchistioides antiguensis* | + | + | − | − |

**Notes.**
+, present; −, absent; v, vestigial.

A mandibular palp is present only in *Palaemon macrodactylus Rathbun, 1902* (Table 2) and *Macrobrachium nipponense* (*De Haan, 1849* (in De Haan, 1833–1850)) (Table 2; Fig. 2C). In both these species the structure of the palp is similar, being three segmented (but see *Fujino & Miyake, 1968* for discussion on variation in this character in *P. macrodactylus*), with the distal segment being more slender and slightly longer than the basal and penultimate segments. Distally-serrulate setae are present (Fig. 1D) on all segments of the palp but most numerous on the distal segment.

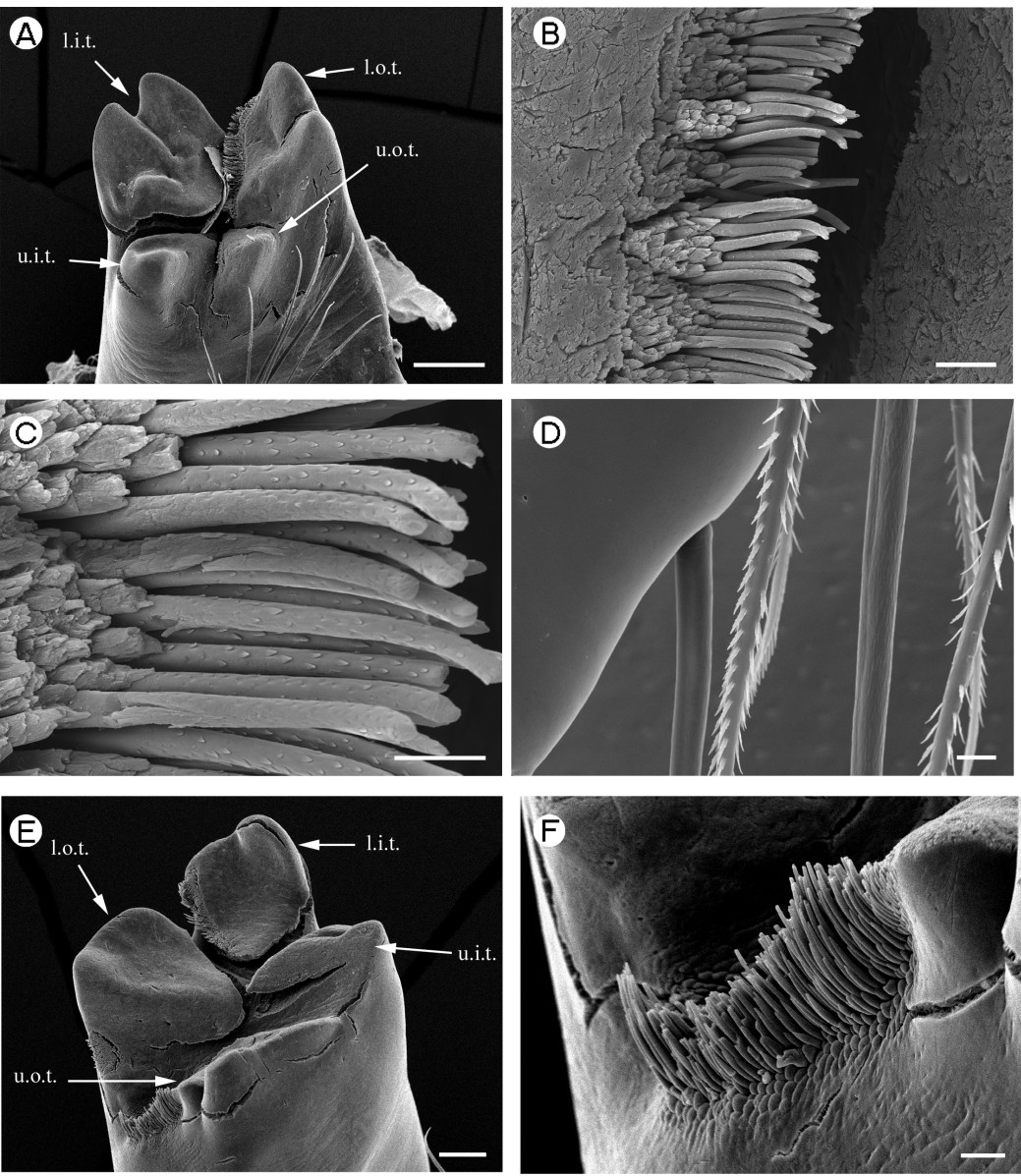

**Figure 1** **Palaemonidae (Palaemoninae):** *Palaemon macrodactylus.* (A) *pars molaris* of right mandible; (B) Type I cuticular structures of right mandible; (C) detail of Type I cuticular structures of right mandible; (D) distally serrulate setae of mandible palp of right mandible; (E) *pars molaris* of left mandible; (F) lateral row of Type I cuticular structures of left mandible. Scale bars indicate 200 μm (A), 100 μm (E), 10 μm (C and D) or 20 μm (B and F). u.o.t., upper outer tooth; u.i.t., upper inner tooth; l.o.t., lower outer tooth; l.i.t., lower inner tooth.

A great diversity of form is present in the *pars molaris*. In all species examined, the *pars molaris* is well developed and ranges from rounded (*P. macrodactylus*, *M. nipponense*, *Periclimenaeus caraibicus* Holthuis, 1951, *H. picta*), oval (*G. elegans*, *Desmocaris bislineata* Powell, 1977, *Euryrhynchus wrzesniowskii* Miers, 1877), slightly squared (*P. pinnophylax*, *Anchistioides antiguensis* (Schmitt, 1924) right) to roughly triangular (*A. antiguensis* left) in cross-section. Most are roughly parallel sided but those of *H. picta*

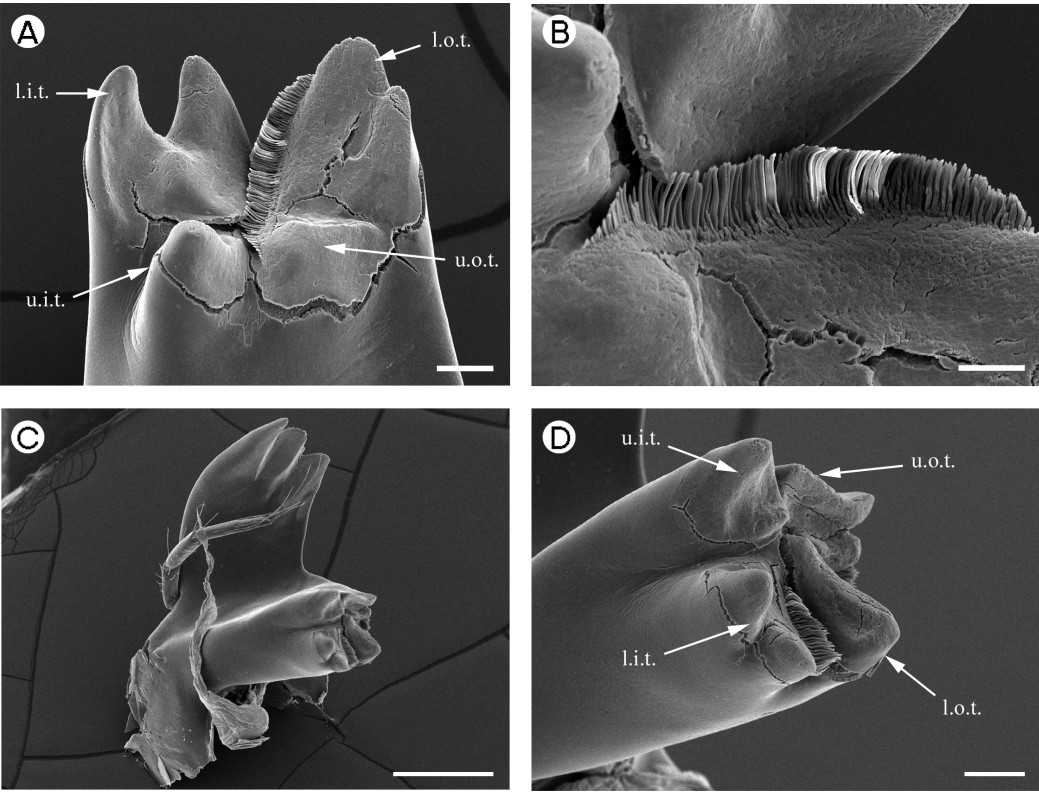

**Figure 2 Palaemonidae (Palaemoninae): *Macrobrachium nipponense.*** (A) *pars molaris* of right mandible; (B) Type I cuticular structures of right mandible; (C) left mandible; (D) *pars molaris* of left mandible. Scale bars indicate 500 μm (C), 100 μm (A and D) or 50 μm (B). u.o.t., upper outer tooth; u.i.t., upper inner tooth; l.o.t., lower outer tooth; l.i.t., lower inner tooth.

and *G. elegans* are strongly curved, that of *D. bislineata* has convex lateral margins and in *A. antiguensis* the *pars molaris* is strongly flared distally. Teeth are present distally on most mandibles (*Palaemon*, Figs. 1A and 1E; *Macrobrachium*, Figs. 2A, 2C and 2D; *Pontonia*, Figs. 3B and 3D; *Anchistioides*, Figs. 7D–7F; *Hymenocera*, Figs. 4E and 4F; *Gnathophyllum*, Fig. 5D), whilst in others these are fused to form lip-like structures (*Euryrhynchus*, Figs. 7A and 7B; *Periclimenaeus*, Figs. 4A–4C) and in *Desmocaris* no teeth are present and the distal end is a ridged plate (Figs. 6A–6B and 6D–6F). The form of the teeth is highly variable, with spine-like teeth being present in *Hymenocera* (Figs. 4E and 4F), a blade like tooth being present in *Gnathophyllum* (Fig. 5D) and more lobate teeth present in the other species. The lobate teeth may be reduced to low mounds or massively produced with the tips entire or bifid as well as all gradations in between. Significant differences in the arrangement and structure of the teeth are also noted between the left and right mandibles. Typically four teeth are present although in some species these are modified such that they are difficult to discern.

In addition to the teeth and cusps mentioned above, the distal end of the *pars molaris* of most mandibles examined here were found to be covered, to a greater or lesser degree, by numerous filamentous structures, which are flexible to semi-rigid and frequently

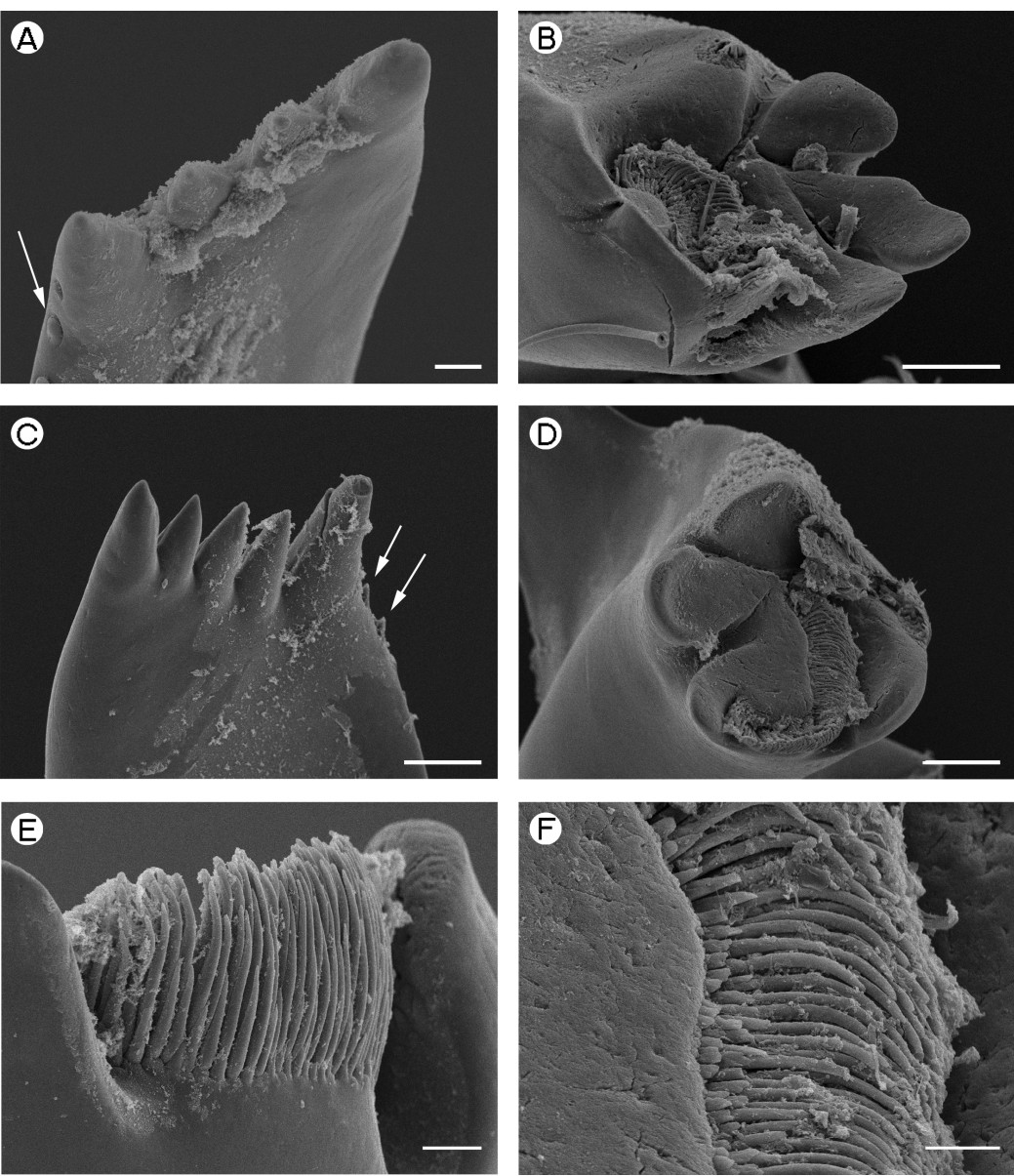

**Figure 3 Palaemonidae (Pontoniinae): *Pontonia pinnophylax.*** (A) *pars incisivus* of right mandible (denticles indicated by white arrow); (B) *pars molaris* of right mandible; (C) *pars incisivus* of left mandible (denticles indicated by white arrow); (D) *pars molaris* of left mandible; (E) Type I cuticular structures of left mandible; (F) Type I cuticular structures of right mandible. Scale bars indicate 100 μm (B and D), 50 μm (C) or 20 μm (A, E and F).

developed into rows (Figs. 1B–1C, 1F, 2A–2B, 3E, 3F, 4B–4C, 4E–4F, 5A–5D, 6A–6F and 7A–7C). The individual filaments do not conform to any described form of seta nor to the definitions of setae in *Watling (1989)* or *Garm (2004)*, in particular lacking a complete basal articulation and a continuous lumen. The arrangement, placement and ultra-structure of these cuticular structures (CS) is highly variable, but can be broadly classified into five types.

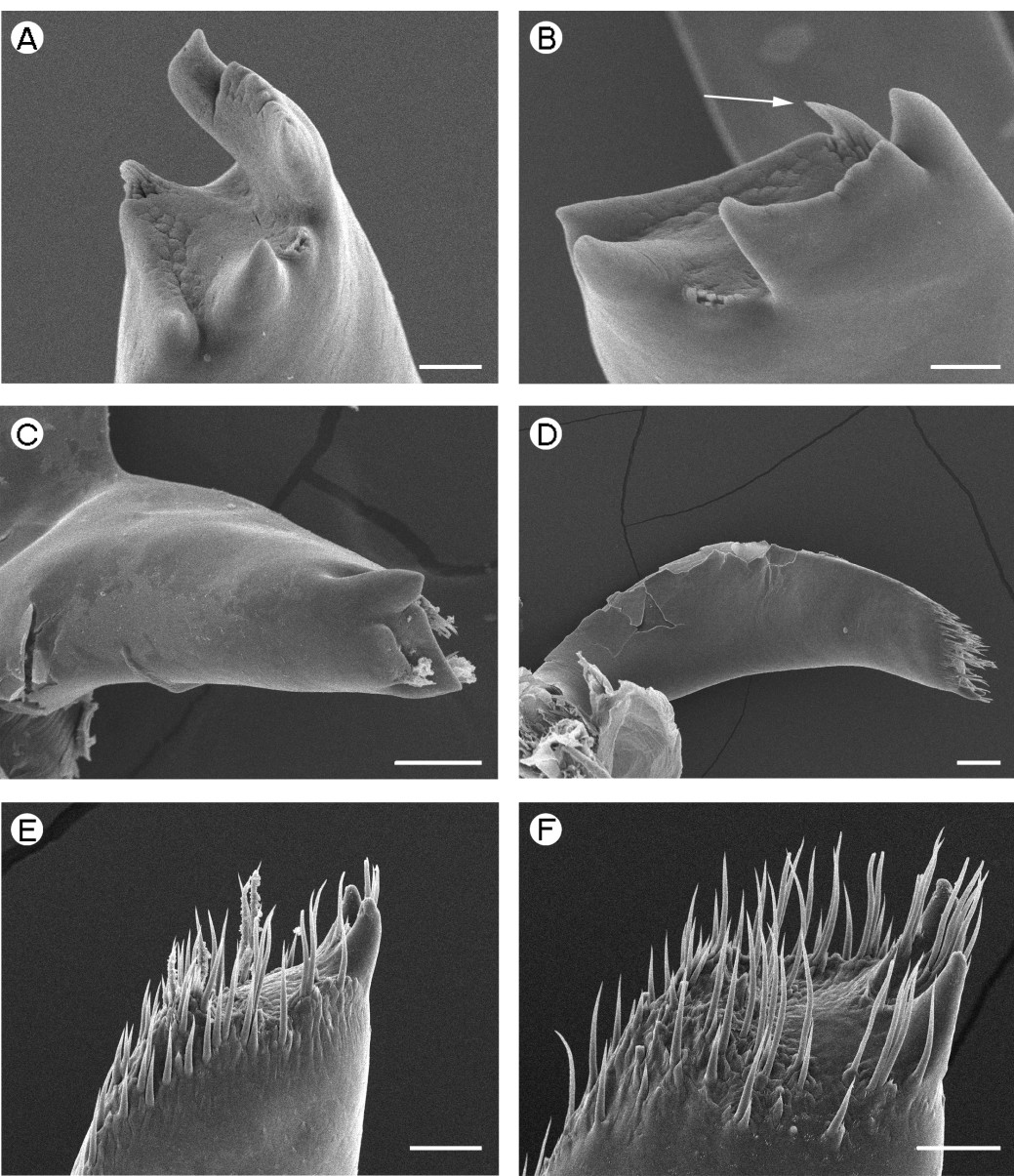

**Figure 4 Palaemonidae: *Periclimenaeus caraibicus* and Hymenoceridae: *Hymenocera picta*.** Palaemonidae (Pontoniinae): *Periclimenaeus caraibicus*, (A) *pars molaris* of right mandible; (B) *pars molaris* of right mandible (spine-like tuft of Type II cuticular structures indicated by white arrow); (C) *pars molaris* of left mandible. Hymenoceridae: *Hymenocera picta*, (D) right mandible; (E) distal end of *pars molaris* of right mandible; (F) distal end of *pars molaris* of left mandible. Scale bars indicate 20 μm (A and B), 100 μm (D), 50 μm (C, E and F).

Type I CS are semi rigid, parallel sided or slightly tapered distally and between 40 and 60 μm long and 3–6 μm wide and tend to form rows. They are found in *Palaemon* (Figs. 1B–1C and 1F), *Macrobrachium* (Figs. 2A, 2B and 2D), *Pontonia* (Figs. 3B and 3D–3F) and *Euryrhynchus* (Figs. 7A–7C). In *Euryrhynchus*, shorter structures are also present (Fig. 7C), but these appear structurally similar to Type I and are herein regarded as the same type.

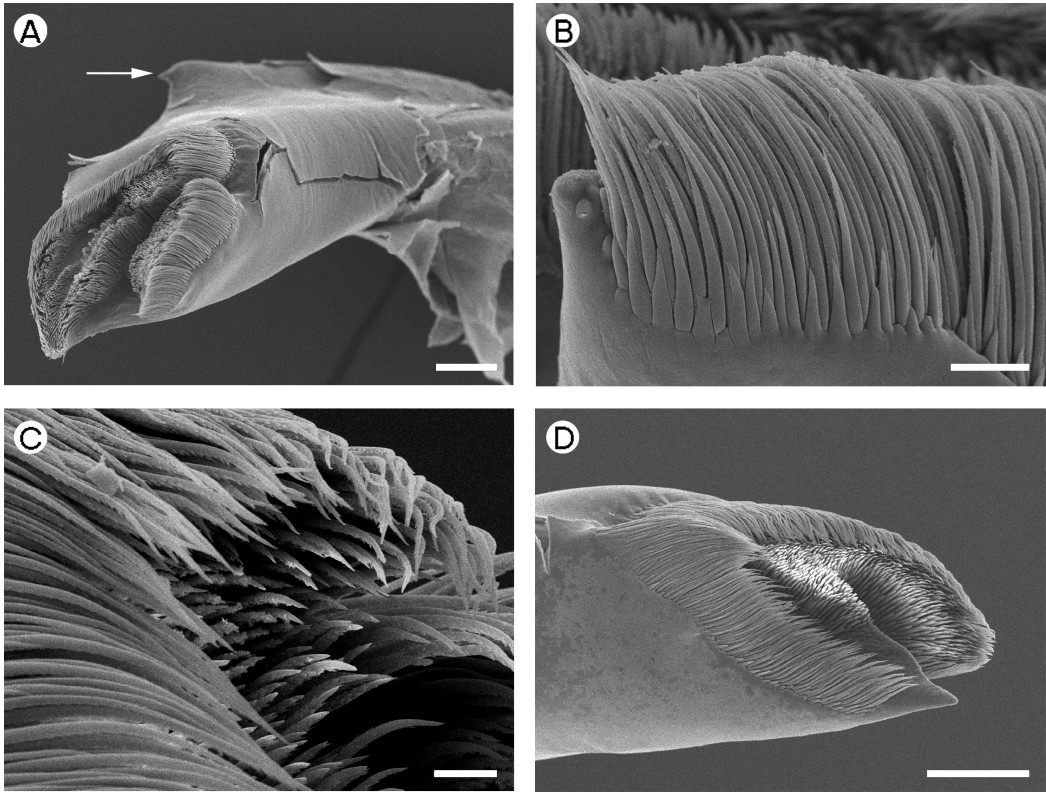

**Figure 5 Gnathophyllidae: *Gnathophyllum elegans*.** (A) *pars molaris* of right mandible (vestigial *pars incisivus* indicated by white arrow); (B) Type III cuticular structures of right mandible; (C) detail of Type III cuticular structures of right mandible; (D) *pars molaris* of left mandible. Scale bars indicate 20 μm (B), 10 μm (C), 100 μm (A and D).

Type II CS are found only in *Periclimenaeus*. These appear more rigid and slightly stouter than Type I structures and form tufts rather than rows (Figs. 4B and 4C).

Type III CS are found in *Gnathophyllum*. They are approximately 60 μm long and 5 μm wide, highly flexible, taper strongly distally with a "feathered" inner margin and have a weak constriction basally (Figs. 5A–5D). They form a dense covering over the entirety of the distal end of the *pars molaris*.

Type IV CS (Figs. 4E–4F) are very similar to Type III differing chiefly in lacking a feathered inner margin and a weak basal constriction. They are exclusively found in *Hymenocera*.

Type V CS are unique to *Desmocaris* and are the most highly modified. They comprise about 12 finger-like projections arising from a basal column (Figs. 6B–6D and 6F).

The details of the positioning and arrangement of the cuticular structures are presented in Table 5 and the figures referred to therein. No cuticular structures were observed on the mandibles of *Anchistioides antiguensis*.

These cuticular structures have been noted in several light microscopy studies or taxonomic descriptions (e.g. *Borradaile, 1917*; *Fujino & Miyake, 1968*; *Felgenhauer & Abele, 1985*; *Storch, Bluhm & Arntz, 2001*; *Fransen, 2006*), where the elements have typically been

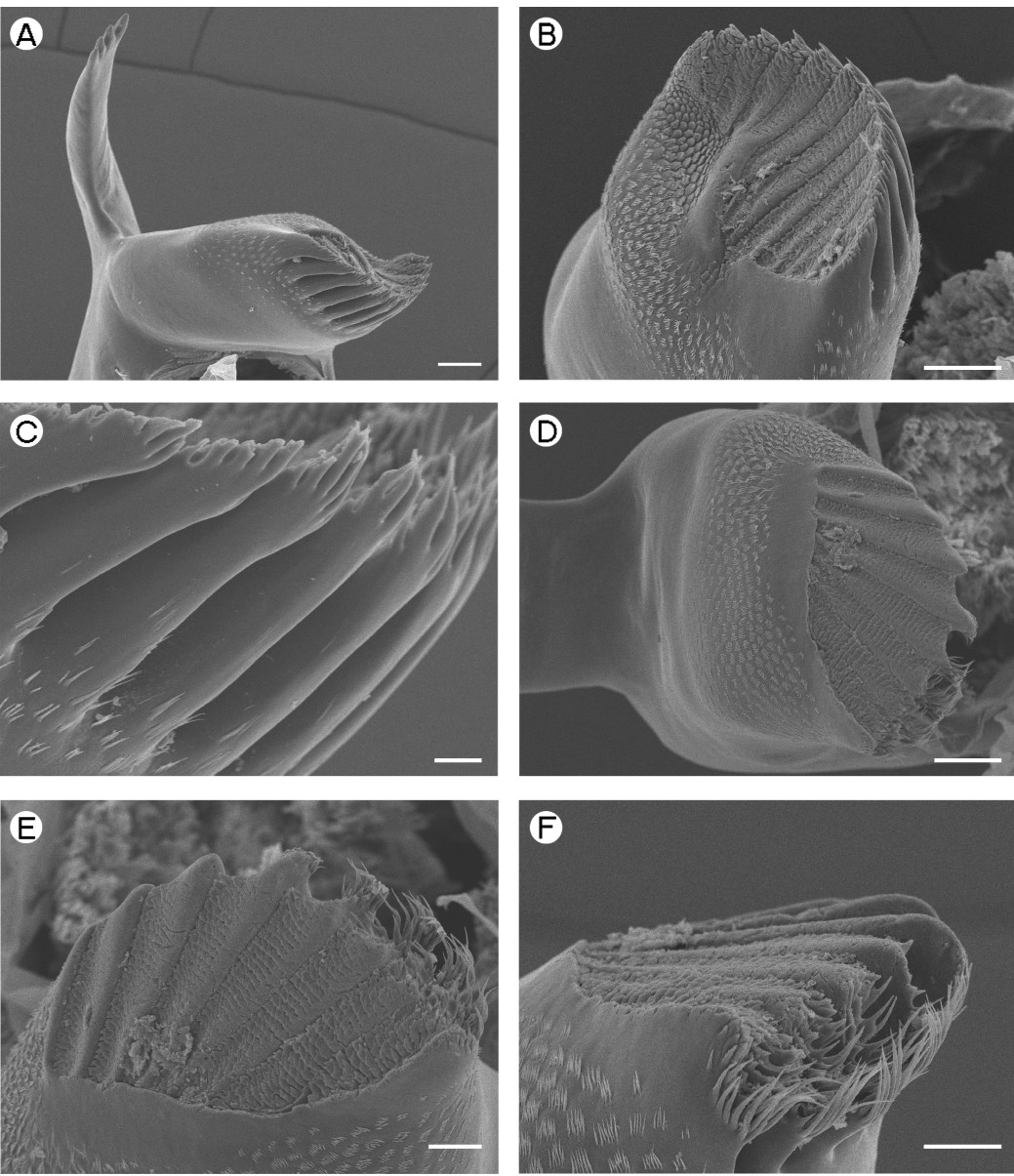

**Figure 6 Desmocarididae: *Desmocaris bislineata*.** (A) Right mandible; (B) *pars molaris* of right mandible; (C) detail of Type V cuticular structures of right mandible; (D) *pars molaris* of left mandible; (E) distal end of *pars molaris* of left mandible; (F) distal end of *pars molaris* of left mandible. Scale bars indicate 100 μm (A, B and D), 20 μm (C), 50 μm (E and F).

referred to as setae or bristles, but no detailed study of these features has been conducted to date. In some species setules are also present on the disto-lateral margins (Figs. 4F, 6B–6C and 6E–6F).

## DISCUSSION

The ecology of palaemonoid shrimp ranges from freshwater to marine habitats and from free-living species to obligate, or loose, associations with a variety of other invertebrates including cnidarians, sponges, echinoderms, molluscs and ascidians. The diversity of

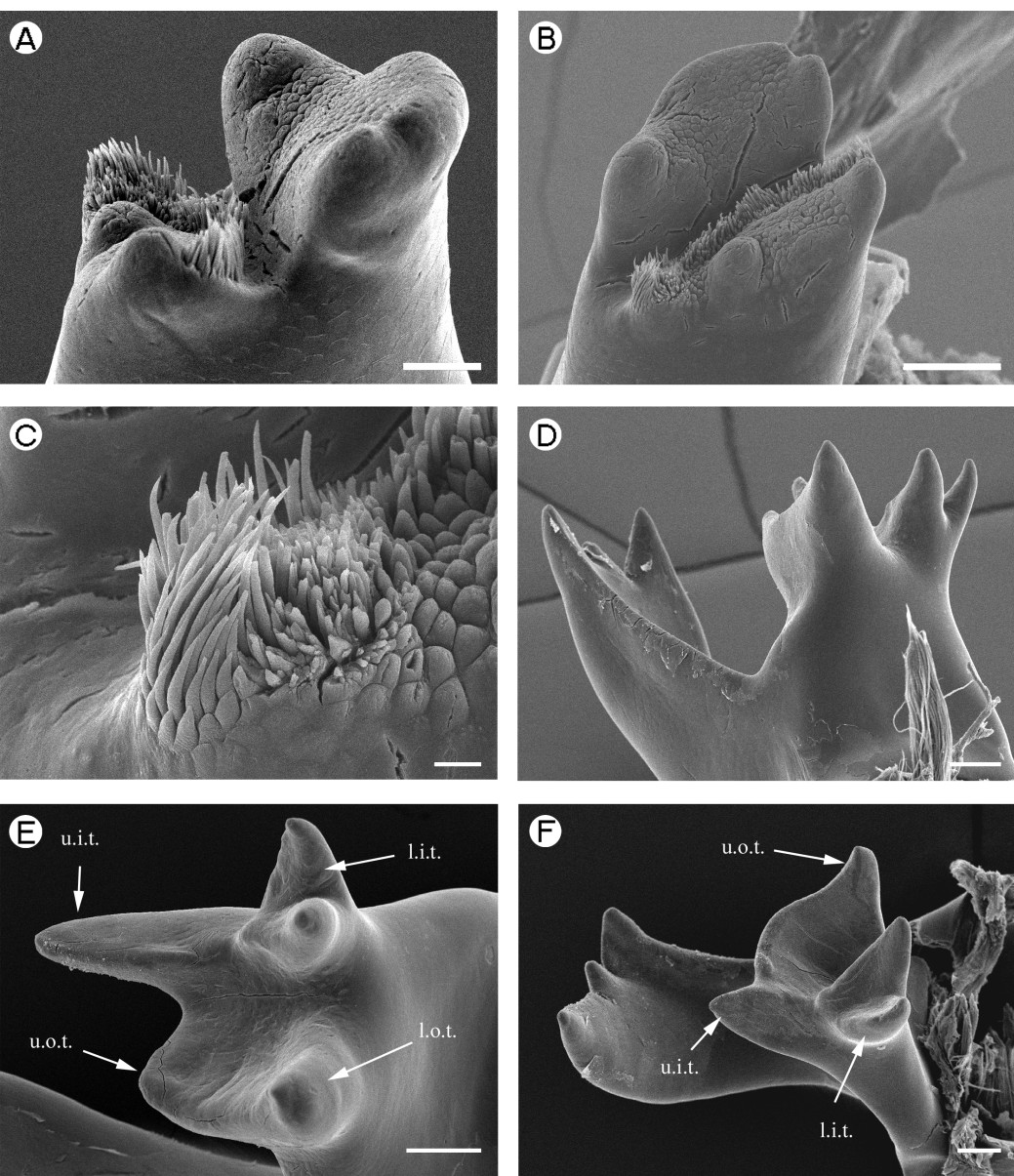

**Figure 7 Euryrhynchidae: *Euryrhynchus wrzesniowskii* and Anchistioididae: *Anchistioides antiguensis*.** Euryrhynchidae: *Euryrhynchus wrzesniowskii*, (A) *pars molaris* of right mandible; (B) *pars molaris* of right mandible; (C) Type I cuticular structures of right mandible. Anchistioididae: *Anchistioides antiguensis*, (D) right mandible; (E) *pars molaris* of right mandible; (F) left mandible. Scale bars indicate 10 μm (C), 100 μm (A, B, D, E and F). u.o.t., upper outer tooth; u.i.t., upper inner tooth; l.o.t., lower outer tooth; l.i.t., lower inner tooth.

lifestyles and feeding strategies within palaemonoid shrimps has resulted in a large range of morphological adaptations, including the mouthparts and they therefore provide an ideal model group to propose hypotheses related to the evolution of these structures. The hypotheses addressed here were that the structure of the mandible should convey information on the species' diet and/or may potentially shed light on the phylogenetic relationships of the taxa.

Ashelby et al. (2015), *PeerJ*, DOI 10.7717/peerj.846

**Table 3** Details of the *pars incisivus* of each species examined.

| | Right form | Anterior margin | Posterior margin | Teeth | Left form | Anterior margin | Posterior margin | Teeth |
|---|---|---|---|---|---|---|---|---|
| *Palaemon macrodactylus* | About twice as tall as wide | Strongly convex | Straight to slightly concave | 3, approximately equal, widely-spaced, triangular. | About twice as tall as wide | Strongly convex | Straight to slightly concave | 4, widely-spaced, triangular, outer teeth slightly larger than inner teeth. |
| *Macrobrachium nipponense* Fig. 2C | Very broad, wider than long in middle portion | Strongly convex | Concave | 3, approximately equal, widely-spaced, triangular. | Very broad, wider than long in middle portion | Strongly convex | Straight | 3, very robust, triangular, anterior most tooth acute, remaining teeth with rounded tip. |
| *Pontonia pinnophylax* Figs. 3A and 3C | Elongate, slender, equal in length to *pars molaris*, strongly curved distally. | Straight, roughly parallel with posterior | Straight, roughly parallel with anterior with seven denticles | 4, triangular, outer teeth larger and broader than inner teeth. | Elongate, slender, equal in length to *pars molaris*, strongly curved distally. | Straight roughly parallel with posterior | Straight, roughly parallel with anterior with five denticles | 5, triangular, acute, posterior-most the largest, remaining teeth approximately equal size. |
| *Periclimenaeus caraibicus* | Slender, ribbon-like, slightly twisted and slightly shorter than pars molaris | Straight roughly parallel with poserior | Straight roughly parallel with anterior | Distally damaged in present specimen, detail from *Holthuis (1951)*: Small acute teeth present distally, about 10 in number. | Laminar in form, slightly curved and slightly shorter than pars molaris. | Convex | Concave | Distal margin broadly rounded, tapering posteriorally, armed with 11 small, acute teeth. |
| *Gnathophyllum elegans* Fig. 5A | Vestigial | – | – | – | Vestigial | – | – | – |
| *Hymenocera picta* | Absent | – | – | – | Absent | – | – | – |

Ashelby et al. (2015), *PeerJ*, DOI 10.7717/peerj.846

Table 3 (*continued*)

| | Right form | Anterior margin | Posterior margin | Teeth | Left form | Anterior margin | Posterior margin | Teeth |
|---|---|---|---|---|---|---|---|---|
| *Desmocaris bislineata* Fig. 6A | Slightly shorter than *pars molaris*, about 3.5 times as long as wide, slightly curved inwards. | Slightly convex | Slightly concave | 4, approximately equal, widely-spaced, triangular | Similar to that of the right mandible, but slightly broader in median part. | Slightly convex | Slightly concave | 4, approximately equal, widely-spaced, triangular |
| *Euryrhynchus wrzesniowskii* | Elongate, slender, about 3.5 times as long as wide, parallel sided, slightly curved inwards. | Straight roughly parallel with poserior | Straight roughly parallel with anterior | 4, widely-spaced, triangular, anterior-most slightly larger than remaining three. | Not examined | – | – | – |
| *Anchistioides antiguensis* Figs. 7D and 7F | Broad, about 3 times as long as wide, slightly twisted. Equal to, or slightly longer than *pars molaris*. | Slightly convex | Slightly concave | 3, widely-spaced, triangular, acute, outer two broader and longer than median tooth. | Broad, about 3 times as long as wide, slightly twisted. Equal to, or slightly longer than *pars molaris*. | Strongly convex | Straight to slightly concave. | 3, widely-spaced, triangular, acute, teeth distally, outer two broader and longer than the median tooth. |

**Table 4** Details of the distal ends of the pars molaris of each species examined.

|  | Right | Left |
| --- | --- | --- |
| *Palaemon macrodactylus* | Quadricuspid (Fig. 1A) | Quadricuspid (Fig. 1E) |
| *Macrobrachium nipponense* | Quadricuspid (Fig. 2A) | Quadricuspid (Figs. 2C and 2D) |
| *Pontonia pinnophylax* | Quadricuspid, with deep concavity (Fig. 3B) | Quadricuspid, teeth flattened (Fig. 3C) |
| *Periclimenaeus caraibicus* | Bifid, 2 acute ridges (Figs. 4A and 4B) | Tricuspid (Fig. 4C) |
| *Gnathophyllum elegans* | Single blade-like tooth (Fig. 5A) | Single blade like tooth (Fig. 5D) |
| *Hymenocera picta* | 2 recurved, spine-like teeth (Fig. 4E) | 2 recurved, spine-like teeth (Fig. 4F) |
| *Desmocaris bislineata* | Ridged (Fig. 6B) | Ridged (Figs. 6D–6F) |
| *Euryrhynchus wrzesniowskii* | 2 lobate ridges (Figs. 7A and 7B) | Not examined |
| *Anchistioides antiguensis* | Quadricuspid (Fig. 7E) | Tricuspid, u.o.t. and u.i.t. fused, wing-like; l.i.t. bifid (Fig. 7F) |

**Notes.**
u.o.t., upper outer tooth; u.i.t., upper inner tooth; l.i.t., lower inner tooth.

Whilst there is considerable variation in the mandible of palaemonoid shrimps noted in the literature, the most common form of mandible across the superfamily is with both a well-developed *pars inscisivus* and *pars molaris*, with a mandibular palp being absent more often than present.

When present, the *pars incisivus* is of fairly constant form, differing only in its robustness and the number of distal teeth, this latter character often being also variable between the left and right mandibles. The *pars incisivus* of *Pontonia* is the most unusual of those investigated here in bearing a row of small denticles on the posterior border. These denticles are also present in most species of the closely related genera *Ascidonia, Dactylonia, Odontonia* but not in *Bruceonia* (see *Fransen, 2002*) but are not described in any other palaemonoid shrimp.

The gross morphology of the *pars molaris* is far more variable between genera than a review of the literature would suggest. This may be partly due to oversights in descriptions or because frequently only one mandible is described and illustrated or simply the limitations of light microscopy. The right and left *pars molaris* in most cases showed significant differences in structure and are often configured such that there is a rough interlocking between the two sides when closed as also noted by *Borradaile (1917)*. More startling is the wide degree of variation and intricacies in design of the cuticular structures. As mentioned, the presence of 'setae' or 'bristles' on the *pars molaris* has been noted in previous studies. However, these cursory mentions do not hint at the diversity in form, placement and arrangement witnessed in comparatively few species examined here.

### Types of mandible and their presumptive function

Based on the form of the mandible herein examined, six types (Types A–F) can be recognised, which appear to relate to feeding mode or diet, although five of these types apply to single species only and the link with specialised food resources would require greater taxon coverage to include other species that share similar diets.

**Table 5** Details of the mandibular cuticle structures of each species examined.

| | Right | Left |
|---|---|---|
| *Palaemon macrodactylus*<br>Figs. 1B and 1C (Right)<br>Fig. 1F (left) | Type I.<br>Well-developed, row<br>along inner margin of l.o.t,<br>feebly developed row on u.o.t. | Type I.<br>In three discrete regions: row<br>along inner margin of l.i.t., small tuft on outer<br>margin of l.o.t., well-developed row on outer<br>margin between l.o.t. and u.o.t. |
| *Macrobrachium nipponense*<br>Figs. 2A and 2B (Right)<br>Fig. 2D (Left) | Type I.<br>Well-developed row along inner margin<br>of l.o.t. and u.o.t. | Type I.<br>Well-developed row along inner margin<br>of u.i.t. and as a small tuft on the outer margin<br>between the l.i.t. and l.o.t. |
| *Pontonia pinnophylax*<br>Figs. 3B and 3F (Right)<br>Figs. 3D and 3E (Left) | Type I.<br>Confined to the concavity in *pars molaris* tip.<br>Arranged in a semicircle, in a rosette-like fashion. | Type I.<br>Well-developed row, curled around outer and<br>inner margin of u.i.t., between l.i.t. and l.o.t. and<br>along posterior margin. |
| *Periclimenaeus caraibicus*<br>Fig. 4B (Right)<br>Fig. 4C (Left) | Type II.<br>Present as a spine-like tuft in position of u.o.t. | Type II.<br>Three distinct tufts one between u.i.t. and l.i.t.,<br>and two on outer margin of l.i.t. |
| *Gnathophyllum elegans*<br>Figs. 5A–5C (Right)<br>Fig. 5D (Left) | Type III.<br>Very well-developed consisting of a single<br>row that curls around to cover the entirety<br>of the distal surface. | As right mandible |
| *Hymenocera picta*<br>Fig. 4E (Right)<br>Fig. 4F (Left) | Type IV.<br>Scattered | As right mandible |
| *Desmocaris bislineata*<br>Figs 6B and 6C (Right)<br>Figs. 6D–6F (Left) | Type V.<br>Arranged into 12 equally spaced ridges giving a<br>scalloped appearance. Median ridges<br>longest and inner ridges notably<br>shorter than outer ridges. | Type V.<br>Ridges broader than those on right mandible,<br>with rounded tips. |
| *Euryrhynchus wrzesniowskii*<br>Figs 7A–7C (Right) | Type I.<br>Arranged in a transverse row. | Not examined |
| *Anchistioides antiguensis* | Absent | Absent |

**Notes.**

u.o.t., upper outer tooth; u.i.t., upper inner tooth; l.o.t., lower outer tooth; l.i.t., lower inner tooth.

*Type A mandible*: Well developed *pars incisivus* and *pars molaris*; *pars molaris* distally cuspidate; with Type I CS; encountered in *Palaemon macrodactylus*, *Macrobrachium nipponense*, *Euryrhynchus wrzesniowskii* and *Pontonia pinnophylax* (Figs. 1–3 and 7A–7C).

*Palaemon macrodactylus* is largely carnivorous with a preference for mysid and amphipod crustaceans (*Sitts & Knight, 1979*; *Siegfried, 1982*; *González-Ortegón et al., 2010*; C Ashelby, 2012, unpublished data). The specific, natural diet of *Macrobrachium nipponense* has not been studied but it is likely that, as with most *Macrobrachium*, it is omnivorous with a tendency towards carnivory (*Jayachandran & Joseph, 1989*; *Mantel & Dudgeon, 2004*; *Short, 2004*). The diet of the congeneric *M. hainanense* (*Parisi, 1919*) is dominated by insect larvae and gastropod molluscs (*Mantel & Dudgeon, 2004*) and a similar diet may be assumed for *M. nipponense*. Although the diet of *Euryrhynchus*

*wrzesniowskii* has not been studied, *Kensley & Walker (1982)* provide some information on the diet of the related *E. amazoniensis Tiefenbacher, 1978*, whilst *Walker (2009)* also gave information on the diet of this species and *E. burchelli Calman, 1907*. Both species feed on a diverse prey range and can be regarded as omnivorous with a preference for live insect larvae. The diet of *Pontonia pinnophylax* is unclear. *Pontonia* inhabit lamellibranch bivalve, gastropod or ascidian hosts (*Fransen, 2002*; *Marin & Anker, 2008*). *Richardson et al. (1997)* concluded that the most likely food sources of *P. pinnophylax* were pseudofaeces (mucous-bound suspended particles rejected as food by the bivalve) or material collecting in the mantle cavity. Similarly, *Aucoin & Himmelman (2010)* observed *Pontonia mexicana* (*Guérin-Méneville, 1855* (in Guérin-Méneville, 1855–1856)) feeding on matter in mucus strings. Gut content analysis has revealed the presence of detrital material, plant material and crustacean exuviae (*Richardson et al., 1997*). Finally, *Kennedy et al. (2001)* concluded that *Pontonia* assimilated similar food to their bivalve hosts based on similar stable isotope carbon measurements.

The hard-bodied, relatively large prey consumed by *Palaemon*, *Macrobrachium* and *Euryrhynchus* would require breaking down prior to ingestion. This suggests the requirement for a grinding mandible and the application of force. The cuspidate nature of the *pars molaris* of the Type A mandible is supportive of such a grinding function. The abraded nature of many of the cuticular structures (particularly evident in Figs. 1B and 1C) also supports this view. It would also be necessary for the shrimp to sense the prey between the mandibles to know what force is being applied to the prey, when the prey had been ground enough to ingest or when exoskeletons or shells of the prey had been broken. This is the presumed function of the Type I CS in the Type A mandible. Type I CS are most similar to microtrichia, which are common in crustaceans, particularly in amphipods (e.g. *Steele & Oshel, 1987*; *Oshel, Steele & Steele, 1988*; *Olyslager & Williams, 1993*; *Wong & Williams, 2009*; *Zimmer, Araujo & Bond-Buckup, 2009*; *Mekhanikova et al., 2012*) and have also been noted in larval decapods (e.g., *Pohle & Telford, 1981*; *Tziouveli, Bastos-Gomez & Bellwood, 2011*). Typically microtrichia are thought to have a sensory function (*Olyslager & Williams, 1993*; *Wong & Williams, 2009*) and usually arise from a socket and terminate in a pore. A socket and pore are not evident in the images used here but this may be due to the abraded nature of many of the structures (see Figs. 1B and 1C).

It is not clear how the presumed diet of *Pontonia* links to this mandible type. Assuming a pseudofaeces or mucus diet is correct, there would not be the same requirement for grinding or mechanosensory structures. Similarly *De Jong-Moreau, Casanova & Casanova (2001)* noted that mandibular structure does not always reflect diet.

Based on examination of stomach content, *Tsurnamal (2008)* suggested that *Typhlocaris ayyaloni Tsurnamal, 2008* feeds on bacterial mats and some small crustaceans. Feeding on bacterial mats may require specialised feeding structures; however, Fig. 2F in *Tsurnamal (2008)* shows a mandible of very similar appearance to that of *Macrobrachium* and *Palaemon* which instead suggests a similar diet. This is further supported by the sensitivity of *Typhlocaris* to vibration (*Tsurnamal, 2008*) which would aid in prey detection. This suggests that small crustaceans may form the greater proportion of the diet of *Typhlocaris*.

Whether cuticular structures are present is not evident from the figures or descriptions in any *Typhlocaris* species.

*Type B mandible*: Well developed *pars incisivus* and *pars molaris*; *pars molaris* distally cuspidate; lacking cuticular structures; only encountered in *Anchistioides antiguensis* (Figs. 7D–7F). It differs from the Type A mandible chiefly through the lack of cuticular structures. The *pars molaris* is also distally flared which is one of the defining characteristics of the family Anchistioididae.

The known species of *Anchistioides* are commonly associated with a variety of shallow water sponges inhabiting the oscula. It may be speculated that they feed either on detritus collected within the osculum of the sponge, other organisms associated with the sponge, the sponge itself, or a combination of these. The only evidence as to the diet of *Anchistioides* was provided by *Wheeler & Brown (1936)* who report the presence of 'worm setae' in the stomachs of two specimens of *A. antiguensis*. The lack of any sensory apparatus may support the idea of this species preying on softer bodies animals which would require less force to break down.

*Type C mandible*: Well developed *pars incisivus* and *pars molaris*; *pars molaris* asymmetrical with 2 acute ridges on right and tricuspid on left; with Type II CS; only encountered in *Periclimenaeus caraibicus* (Figs. 4A–4C). There is a considerable degree of variation in the mouthparts of *Periclimenaeus* spp. reported in the literature and thus this type of mandible may not be standard for the genus as a whole. In literature (see *Holthuis, 1951*; *Holthuis, 1952* for examples), variation in the development of the *pars incisivus* is noted as well as variation in the development or presence of cuticular structures but this latter difference may again be attributable to oversight in the descriptions and figures due to difficulties observing this feature under light microscopy. The ecological and perhaps phylogenetic significance of variation in features of the mandible amongst *Periclimenaeus* species warrants further investigation.

*Ďuriš et al. (2011)* report that *Periclimenaeus caraibicus* feeds on the host sponges, noting the presence of spicules in the stomach and that the shrimp takes on the colour of the host sponge through assimilation of the sponge's pigments. The form of the mandible witnessed here is also suggestive of a specialised diet. The multidentate, serrated form of the *pars incisivus* would aid in the shredding of sponge fragments, whilst the acute nature of the ridges of the right *pars molaris* may also aid in tearing. The sponge fragments may then be transferred into the groove of the right *pars molaris* into which the teeth of the left *pars molaris* can interlock to grind the sponge down. The groove may also help align unbroken spicules such that they enter the mouth in the correct orientation. The function and placement of the Type II CS in this mandible is difficult to explain. They appear similar in form to Type I CS and may therefore also be assumed to have a similar sensory function but their placement in discrete tufts may suggest a slightly different function. It is speculated that these tufts of cuticular structures are the vestiges of those found in *Pontonia* (see Figs. 3E and 3F) and that they only have limited functionality.

Sponge feeding cannot be presumed to be a generalised diet for *Periclimenaeus*, as some other members of this genus are associates of compound ascidians (*Fransen, 2006*) and

so presumably have different feeding ecology which may be reflected in the form of their mandible, as discussed above.

*Type D mandible*: *Pars incisivus* strongly reduced to vestigial spine-like process; *pars molaris* with single blade-like tooth distally; with Type III CS; only encountered in *Gnathophyllum elegans* (Fig. 5). Type D mandibles are highly modified and display a number of unusual features, most notably the reduction of the *pars incisivus* and the dense covering of Type III CS.

Little information is available on the diet of *Gnathophyllum*. Both *Wickler (1973)* and *Bruce (1982)* speculate that *Gnathophyllum* are predatory on echinoderms, however this hypothesis has not been confirmed. However, the highly modified form of all their mouthparts is suggestive a specialised food resource. During feeding, shrimps use the anterior mouthparts (maxillae and maxillipeds) to hold and manipulate food (*Bauer, 2004*). The operculate, calcified nature of the anterior mouthparts may not be able to manipulate food in the same way as the more flexible mouthparts found in most of the other genera examined here. The strongly reduced *pars incisivus* is suggestive that there is not a requirement for tearing or shredding of food items and the lack of a grinding surface on the *pars molaris* indicates that there is no requirement for breaking down food. Furthermore, the mandibles of *Gnathophyllum* are exceedingly small in relation to the body size of the shrimp and would be unlikely to be able to deal with large food items. Finally, the Type III CS appear highly flexible and cilia-like. These various adaptations would suggest that rather than large food items, *Gnathophyllum* feed on small particulate matter, mucus or fluids or perhaps echinoderm tube-feet and that the Type III CS are involved in movement of these food resources.

Although some species of Gnathophyllidae are commensal with echinoderms (*Bruce, 1982*), *Gnathophyllum elegans* is considered free living. However, *Gnathophyllum* spp. do seem to form loose associations with echinoderms (S De Grave, pers. obs., 2014) and *Bruce (1982)* reports that *G. americanum* (*Guérin-Méneville, 1855* (in Guérin-Méneville, 1855–1856)) has been observed using its outer maxillipeds to browse on the extended papulae on the dorsal surface of asteroids. This, combined with the modifications to the mandible further supports the idea that *Gnathophyllum* feed on mucus or mucus entrapped particles, as has also been suggested by *Bruce (1982)* for some other echinoderm associates such as *Zenopontonia rex* (*Kemp, 1922*) (as *Periclimenes imperator Bruce, 1967*), *Lipkemenes lanipes* (*Kemp, 1922*), *Z. soror* (*Nobili, 1904*) and *Periclimenes pectiniferus Holthuis, 1952*.

*Type E mandible*: *Pars incisivus* absent; *pars molaris* bearing two recurved spine-like teeth distally; with Type IV CS; encountered only in *Hymenocera picta* (Figs. 4D–4F).

This type of mandible is differentiated from the Type D mandible through the complete absence of the *pars incisivus*, the presence of two recurved teeth on each mandible rather than a single blade-like tooth, and by the form and arrangement of the cuticular structures. As in the Type D mandible the *pars molaris* lacks a grinding surface.

*Hymenocera* and *Gnathophyllum* are so similar in the form of the mandible as well as their other mouthparts (a factor that has lead to their previous inclusion in a single family)

that it would be reasonable to assume a similar diet. However, *Wickler (1973)* noted that *Hymenocera* feed exclusively on starfish, particularly *Nardoa* and *Linkia* spp. piercing the epidermis with their first pereiopods before extracting internal tissues.

The sparse arrangement of cuticular structures would also not be as effective at moving mucus or particles as those in the Type D mandible of *Gnathophyllum*. It seems likely, therefore, that the Type E mandible is a further development of the Type D mandible in response to a dietary switch in *Hymenocera* (or its ancestors) from merely removing mucus from the echinoderms to actually predating on them. The paired teeth of the right *pars molaris* apparently interlink with those of the left and may take on the slicing role normally attributed to the *pars incisivus*.

*Type F mandible*: Well developed *pars incisivus* and *pars molaris*; *pars molaris* distally flattened and ridged; with Type V CS; only encountered in *Desmocaris bislineata*.

Type V CS are the most highly developed of all the cuticular structures noted in this study. They in turn dictate the form of this mandible type as the finger-like projections together form the ridged surface of the *pars molaris*. They appear to be flexible and may be regarded as shorter versions of the cilia-like Type III CS. A particulate or detritivorous diet may therefore be expected. This is consistent with the information provided by *Powell (1977)* who states that 'normal feeding activity involves exploration of the surface of dead leaves etc., . . . most of the food probably consists of fine particles, . . . , captive shrimps recoil from contact with live animals such as naidid oligochaetes and chironomid larvae; however they eagerly consume dead ones and therefore do not seem to be restricted to microphagy.' Although a strong *pars incisivus* is present for initial tearing, the Type F mandible does not have obvious grinding function and it is unclear how these carrion prey items would be broken down prior to ingestion. Another possible function for the elaborate arrangement of cuticular structures in this mandible type is that they may help to filter particular matter.

## Systematic considerations

The form of the mandible was considered by *Thompson (1967)* to be of significant importance in the phylogeny of the Caridea, with the ancestral state considered to be a fused *pars molaris* and *pars incisivus*, combined with a 3-segmented palp. Indeed, the recognition of several families, including some incorporated in this study, has partially been justified by the form of the mandible. The ridged nature of the *pars molaris*, which is presumed to be a primitive feature (*Sollaud, 1911*; *Borradaile, 1917*) is one of the characters used to define the family Desmocarididae (*Borradaile, 1915*; *Powell, 1977*) and the presence of a distally flared molar process of the mandible is one of the defining characteristics of the family Anchistioididae (*Chace, 1992*). However, *Fransen & De Grave (2009)* concluded that whilst the form of mandible is of considerable value in the identification of carideans, its phylogenetic significance at the family level is uncertain. The inclusion of relatively few species in this study, encompassing less than 1% of palaemonoid diversity, albeit from the majority of palaemonoid families, will not uncover the complete range of forms of the mandible likely to be found in this group, meaning that the results of this study should be regarded as indicative rather than absolute. Furthermore, the analysis of a single character

in isolation cannot hope to resolve systematic relationships, rather an integrative approach, including novel characters and possibly also molecular data is advised (*Li et al., 2011*). Nevertheless some preliminary observations on the structure of the mandible in relation to currently accepted phylogenies can be made.

The six mandibular types proposed here do not reflect currently accepted relationships within the Palaemonoidea. As many of the groupings are based on single taxa they may actually imply species specific differences or, perhaps reflect over-splitting of mandibular types in this study.

The genera *Palaemon* and *Macrobrachium*, both currently assigned to the Palaemoninae, have the same general structure of the mandible (Type A); however, the other genera with this form of mandible are more difficult to explain from a phylogenetic point of view. *Pontonia* shares a greater affinity to *Gnathophyllum*, *Hymenocera* and *Periclimenaeus* (*Mitsuhashi et al., 2007*; *Bracken, De Grave & Felder, 2009*; *Gan et al., 2015*) than to *Palaemon* or *Macrobrachium* whilst *Euryrhynchus*, considered to be an ancient lineage (*De Grave, 2007*), represents a sister group to *Desmocaris* (see *Bracken, De Grave & Felder, 2009*). *Palaemon* and *Macrobrachium* both also possess a mandibular palp. The traditional view of the mandibular palp is that the presence of a three segmented mandibular palp represents the primitive condition in Caridea (*Thompson, 1967*) with a reduction in the number of segments and subsequent loss in more derived lineages. However, the presence or absence of a mandibular palp has been demonstrated to convey very limited phylogenetic information and is not a consistent character in Palaemonidae, varying even within a species (*Ashelby et al., 2012*; *De Grave & Ashelby, 2013*).

Although classified into two different mandible types here (Type D and Type E), the mandibles of *Gnathophyllum* and *Hymenocera* are linked through the reduction of the *pars incisivus*, a feature that is variable in the gnathophyllid genus *Gnathophylloides* (see *Chace & Bruce,*). *Mitsuhashi et al. (2007)*, *Bracken, De Grave & Felder (2009)* and *Gan et al., 2015*, based on a molecular phylogeny, demonstrated that Hymenoceridae and Gnathophyllidae represent a derived lineage within the Pontoniinae. The mouthparts present many of the definitive morphological characters of this lineage. The gradual reduction of the *pars incisivus* witnessed in the Gnathophyllidae and Hymenoceridae is also a feature demonstrated in several Pontoniinae taxa indicating the potential plasticity of this character within the subfamily. Reduction of the *pars incisivus*, although to a lesser degree, is also evident in Fig. 8A in *Bruce & Short (1993)* of *Calathaemon holthuisi* (*Strenth, 1976*) (ex-Kakaducarididae, now Palaemonidae). A gradual reduction of the *pars incisivus* at family level is indicated by *Burukovsky (1986)* with Gnathophyllidae being intermediate in form between Palaemonidae and Crangonidae. However, these latter families, and the Eugonatonotidae in which the *pars incisivus* is also absent, are not closely related (*Mitsuhashi et al., 2007*; *Bracken, De Grave & Felder, 2009*; *Li et al., 2011*) suggesting that the loss of the *pars incisivus* has occurred independently several times in the evolution of the Caridea.

This study has demonstrated that the form of the mandible is much more complex than previously thought. The traditional view that the *pars molaris* is used solely for the

grinding of food seems a gross oversimplification and in some species (e.g. *G. elegans*, *H. picta*) the arrangement and form of the teeth would suggest that it does not grind at all. The form and arrangement of cuticular structures at the distal end of the *pars molaris* shows a particularly high degree of variation. The five types of cuticular structures recognised in this study are presumed to have different functions related to food sources, which is contrary to the findings of *Storch, Bluhm & Arntz (2001)* who found no link between the morphology of the mouthparts and food items.

Some evidence of evolutionary relationships is conveyed through the broad structure of the mandible but the detailed structures witnessed in this study do not reflect the evolutionary relationships in the Palaemonoidea suggested by previous phylogenetic reconstructions (*Mitsuhashi et al., 2007*; *Bracken, De Grave & Felder, 2009*; *Li et al., 2011*). This preliminary study thus suggests that the structure of the mandible is more related to function in relation to diet, than evolutionary relationships. With such a diversity of lifestyles represented by the Palaemonoidea, particularly within the subfamily Pontoniinae, further studies including many other genera are however required to fully unravel the diversity of mandible morphology within the superfamily.

## ACKNOWLEDGEMENTS

John Short and an anonymous reviewer made useful comments and corrections.

### Funding

Some of the photos used in the present study were taken during a summer OUMNH EPA/Cephalosporin funded internship. The work of the first author was partially supported by Unicomarine Ltd. The funders had no role in study design, data collection and analysis, decision to publish, or preparation of the manuscript.

### Grant Disclosures

The following grant information was disclosed by the authors:
OUMNH EPA/Cephalosporin.
Unicomarine Ltd.

### Competing Interests

Christopher W. Ashelby is an employee of APEM Ltd. Magnus L. Johnson is an Academic Editor for PeerJ.

### Author Contributions

- Christopher W. Ashelby conceived and designed the experiments, performed the experiments, analyzed the data, wrote the paper, prepared figures and/or tables, reviewed drafts of the paper.
- Sammy De Grave conceived and designed the experiments, performed the experiments, analyzed the data, contributed reagents/materials/analysis tools, wrote the paper, reviewed drafts of the paper.

- Magnus L. Johnson analyzed the data, wrote the paper, reviewed drafts of the paper.

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
