# Peer review of "Preliminary observations on the mandibles of palaemonoid shrimp (Crustacea: Decapoda: Caridea: Palaemonoidea)"

_PeerJ, doi:10.7717/peerj.846_

## Round 0.1 · original submission · Minor Revisions

I enjoyed your manuscript, in addition to reviewer two I have some minor comments:
1. Please introduce abbreviations when first used.
2. Please include the nominal authority for binomia on their first usage.

Reviewer 1 ·

Basic reporting

No Comments

Experimental design

No Comments

Validity of the findings

No Comments

Additional comments

Excellent paper. I corrected few typos in the text file.

Annotated reviews are not available for download in order to protect the identity of reviewers who chose to remain anonymous.

·

Basic reporting

This is the first significant study of the ultrastructure of palaemonoid shrimp mandibles using SEM. The paper is highly informative and well written. There are relatively few typographic or grammatical errors.
The literature has been appropriately referenced apart from a few redundant references in the reference list which were not mentioned in the text.
The SEM figures are of a high standard although there appears to a problem with labeling for Figs 7A-C (see also annotated manuscript). Fig 7A looks like the right mandibular molar process of a different species compared to 7B-C.

Experimental design

The scope and objectives of the study are clearly stated in the introduction. The methodology is also well explained and appears sound.

Validity of the findings

The findings of the study adequately address the objectives stated in the introduction. The six mandible types recognized for the Palaemonoidea have been well defined. The hypothesized relationships to feeding mode or diet appear to be sensible but have also been appropriately qualified due to the limited number of species studied. The conclusion that the structure of palaemonoid mandibles is more related to function in relation to diet rather than an indication of evolutionary appears well justified based on the results of the study.

Additional comments

All suggestions/corrections are included on the annotated manuscript.

---

## Round 0.2 · Minor Revisions

Dear authors,

Thank you for your revised manuscript. I noticed some of the newly added citations have not been added into the reference list. Prior to acceptance, could you do one final check of all in-text citations, and the reference list, to ensure they match.

---

## Round 0.3 · accepted · Accept

Thank you for your swift response to the review comments, and I hope you choose PeerJ as your publication outlet in the future.